# Atrazine Desorption Mechanism from an Hydrated Calcium Montmorillonite—A DFT Molecular Dynamics Study

**DOI:** 10.3390/ijms25031604

**Published:** 2024-01-27

**Authors:** Quentin Desdion, Fabienne Bessac, Sophie Hoyau

**Affiliations:** 1Laboratoire de Chimie et Physique Quantiques, Université de Toulouse, Université Paul Sabatier, Toulouse III, CNRS (UMR 5626), 118 Route de Narbonne, F-31062 Toulouse, France; qdesdion@irsamc.ups-tlse.fr (Q.D.); sophie.hoyau@irsamc.ups-tlse.fr (S.H.); 2Ecole d’Ingénieurs de Purpan, Université de Toulouse, Toulouse INP, 75 Voie du TOEC, BP 57611, Cedex 03, F-31076 Toulouse, France

**Keywords:** atrazine, clay, desorption, hydration, DFT, molecular dynamics

## Abstract

Atrazine is one of the most widely used herbicide molecules in the triazine family. Despite its interdiction in the European Union in 2004, atrazine and its main degradation products remain among the most frequently found molecules in freshwater reservoirs in many European Union countries. Our study aims in obtaining insight into the desorption process of atrazine from the main soil absorbent material: clay. Constrained Molecular Dynamics simulations within the Density Functional Theory framework allow us to obtain a free energy desorption profile of atrazine from a Ca^2+^-montmorillonite surface. The results are interpreted in terms of atrazine inclination to the clay surface and moreover, in terms of hydration states of the cations present in the clay interlayer as well as the hydration state of the atrazine. The desorption mechanism is driven by atrazine alkyl groups and their sizes because of dispersion stabilizing effects. The highest barrier corresponds to the loss of the isopropyl interaction with the surface.

## 1. Introduction

The sorption of pesticides and, more broadly, of organic contaminants, is an important factor to study in soil and water contamination processes, as well as in the development of methods for removing pollutant molecules from these environmental compartments. A wide range of studies concerns the investigation of adsorption and desorption of pesticides from soils or absorbent materials such as biochar, clays or activated carbon [1,2,3]. Among the many pesticide molecules, atrazine (see Figure 1) is the active ingredient in herbicides.

It is one of the most widely used herbicide molecules in the triazine family. Atrazine is a selective herbicide that controls the development of broadleaf weeds and grasses in major crops such as maize, soybean, citrus fruits and sugarcane. Atrazine and its main degradation products are of concern due to their potential endocrine and carcinogenic activity [4]. The introduction of atrazine dates back to the late 1950s. Since its introduction, atrazine has heavily been used. Thus, atrazine and its metabolites are widely present in the environment, contaminating ground and surface water, entering the atmosphere and suspected of persisting for decades in soils. In Europe, the use of atrazine was curtailed in the early 1990s due to excessively high concentrations found in drinking water [5]. In 2004, the European Union finally banned its use. However, atrazine remains one of the most frequently found molecules in freshwater reservoirs in many European Union countries, including France and Germany. Clays in soils play an important role in immobilizing many organic contaminants, such as atrazine and its metabolites [6]. Thus, atrazine has extensively been studied using experimental and theoretical resources and techniques. For instance, G. Abate et al. conducted adsorption and desorption experiments. They described the modification of the clay minerals, vermiculite and montmorillonite, by intercalating Fe(III) polymers with the aim of removing atrazine and its metabolites: deethylatrazine, deisopropylatrazine, and hydroxyatrazine from aqueous solution. The results of the desorption studies in water allow for the depiction of the following affinity order for K^+^ homoionic montmorillonite clay: hydroxyatrazine > deisopropylatrazine ≈ atrazine ≈ deethylatrazine [7]. In 2011, N. D. Jablonowski et al. wrote a short commentary article to alert about the use of atrazine regarding its persistence and toxicity and those of its metabolites [4]. A wide variety of theoretical and experimental studies have been carried out on atrazine, some of which combine theory and experience. We will comment here on some of the most recently published. Y. Han et al.’s computer simulation results aim for serving as a guideline for experimental design in a study to detect the trace level of atrazine residue in the environment and crops [8]. In another study similar to that of Y. Cheng et al., DFT calculations provided an interesting tool to reinforce the experimental results. Here, electrostatic potential showed evidence that the good adsorption performance of nitrogen (N)-doped cellulose biochar for atrazine mainly depends on chemisorption, and π−π electron donor-acceptor interaction [2]. In a work carried out by the V. N. Freire group, the binding of atrazine to human serum albumin was characterized. Describing those interactions is clinically relevant for future pharmacological and toxicological studies. The binding was investigated by using fluorescence spectroscopy and performing simulations using molecular docking, classical molecular dynamics and quantum biochemistry based on density functional theory (DFT) [9]. Similarly, Y. Vieira et al. explored adsorption mechanisms of two pesticides, atrazine and diuron, onto a model of activated carbon and used conceptual DFT descriptors to analyze, confirm or complete their experimental results. The atrazine molecule changes its interaction orientation from inclined to parallel to the activated carbon surface, resulting in higher electrostatic repulsion, reflecting on its low increase in adsorption capacity as the temperature increases [3]. In 2021, D. Tunega and coworkers worked on the stability of atrazine–smectite intercalates using DFT simulations and implementing experimental data. Smectites of this study were montmorillonite and beidellite. The calculations revealed an arrangement of the atrazine in the interlayer space parallel to the surface of both smectites. The stability of beidellite structures was higher. Moreover, the authors showed that organically modified smectite increased the fixation of atrazine [10].

Our study is focused on the desorption of atrazine from a Ca^2+^–montmorillonite surface (Mont) in the presence of water. Previously, we published the results of simulations on the desorption of a fungicide, the fenhexamid, from the same hydrated surface [11]. In this work, we first present the results: we will describe the free energy desorption profile before we discuss the free energy changes in terms of atrazine inclination to the Mont surface, dispersion effects and hydration state of both the cations and of the atrazine molecule. In the Materials and Methods part, we will present the model and the methods we used to investigate the desorption of the atrazine from Mont.

## 2. Results and Discussion

### 2.1. Description of the Desorption Profile

In Figure 2, the free energy desorption profile of Atra from surf is depicted. Five singularities are noticeable and labeled P1, M1, P2, M2 and P3. The zones delimited by the red horizontal lines enclose the structures considered as belonging to the minima and to the maxima. P1, P2 and P3 are free energy minima separated by two barriers evaluated from the smoothed curve at 1.5 and 11.5 kcal·mol^−1^, a low and a high barrier, respectively. Taking the minimum free energy point for a well and the maximum free energy point for a maximum, these barriers are worth 2 and 13 kcal·mol^−1^, respectively. M1 and M2 are maxima along the reaction coordinate ξ ($\xi_{M1}=3.88$ Å and $\xi_{M2}=5.35$ Å). P1 and P2 are quasi isoenergetic: P1 and P2 are centered on ξ values of 3.41 Å and 4.24 Å, respectively. In addition, P3 centered on ξ=5.56 Å is close to M2 in free energy (≈0.5 kcal·mol^−1^). After P3, the free energy increases with ξ without anymore clear singularity. To analyze this part of the profile, we define three ξ ranges 0.4 Å wide centered on 6.0, 7.0 and 8.0 Å.

In the Appendix A, information is detailed confirming the liquid phase of water all along the desorption area (see Appendix A). Moreover, water molecules are not directly in contact with the Mont surface and the hydration of the surface stays similar all along the desorption process (see Appendix A).

The compensating cations, in our case Ca^2+^, are initially located close to surf (<0.5 Å) and in the center of rings formed by six Ob atoms, thus they lie above an O sheet isomorphic substitution. Ca1 is defined as the cation interacting with Atra. The starting interaction mode is bidentate and involves Cl, the chlorine atom linked to C6 carbon atom and N1, the nitrogen atom of the triazine cycle bonded to C6 and the closest to the isopropyl group (see Figure 1). This complexation site corresponds to the lowest-energy isomer in the gas phase for the Atra-Ca^2+^ complex at the PBE level (unpublished results).

### 2.2. Interaction of Atrazine with the Ca1 Cation

To analyze the complexation site of Ca1 on the atrazine molecule, we will refer to Figure 3a,b. In Figure 3a, the normalized Ca1-N1 distance distribution shows that for P1, M1 and P2, the interaction involves N1, a nitrogen atom of the triazine cycle. From M2 and larger ξ values, the Ca1-N1 interaction is lost. Furthermore, in Figure 3b, the normalized Ca1-Cl distance distribution is presented. For P2, this figure evidence the interaction of Ca1 with Cl, the chlorine atom of Atra (Ca1-Cl ≈3.2 Å), implying a bidentate complexation site between Ca1 and Atra. For P1 and M1, the Ca1-Cl distance is longer (≈3.7–3.8 Å) leading to a monodate interaction site. From M2 and larger ξ values, all Ca1-Atra interactions are lost: for Ca1-N1 as well as the Ca1-Cl. The difference between P1 and P2, which are energetically quasi degenerated, concerns the interaction site; monodentate for P1 and bidentate for P2.

### 2.3. Atrazine Inclination with Respect to the Surface

We define the angle $\hat{XClN3}=\theta$ to follow the evolution of the atrazine triazine cycle inclination with respect to the surface (see N3 in Figure 1). The coordinates of the ghost atom *X* are $\left(x_{N3};\;y_{N3};\;z_{Cl}\right)$. For θ values close to zero, the triazine cycle is almost parallel to the Mont surface. When θ is positive, $z_{N3}>z_{Cl}$, whereas when θ is negative, $z_{N3}<z_{Cl}$ (see Figure 4a). In Figure 4b, θ normalized distribution is given for each singularity of the profile and also for the ξ values 6.0, 7.0 and 8.0 Å. From P1 to M2, θ increases.

As we can see in Table 1, the mean value of θ changes from 7.1^∘^ in P1 to 48.7^∘^ in M2. However, in M2, the distribution presents two peaks: one around 30^∘^ and the other one around 60^∘^. P3, the next singularity, shows a single peak which mean value (56.7^∘^) is also close to 60^∘^ as in M2. For the ξ values around 6.0, 7.0 and 8.0 Å, the mean θ value decreases from 40.3^∘^ to 27.1^∘^. Around 6.0 and 7.0 Å, the angle distributions present two peaks centered on 0^∘^, 50^∘^, and on 20^∘^, 40^∘^, respectively. Around 8.0 Å, the distribution is wide from 0 to 50^∘^, with a mean value of 27.1^∘^. In the beginning of the desorption process, the Atra triazine cycle is almost parallel to the surface. θ oscillates around 0^∘^. When it is negative, the N3 nitrogen atom is closer to the surface than Cl. Early in the process, negative angles can be found only in the P1 and M1 singularities. The increase in the θ angle from P1 to P3 moves together with a loss of dispersion effects due to the detachment of the functionalized triazine cycle from the surface but a conservation of the Cl–surface dispersion interaction. Thus, the largest variation in θ is observed between P2 ($\bar{\theta }=20.8^{\circ }$) and M2 ($\bar{\theta }=48.7^{\circ }$), which corresponds to the highest free energy barrier in the profile, i.e., 11.5 kcal·mol^−1^. Beyond P3, those dispersion effects are also lost and θ angle varies more. The evolution of the atrazine inclination with respect to the surface, described by θ, is correlated to the variation of atrazine and Ca1 hydration states.

### 2.4. Dispersion Effects and Atrazine and Cation Hydration States

In this part, we will discuss the hydration states of the atrazine molecule and of the Ca^2+^ cations. In Table 2, the minimum, maximum and mean numbers of water molecules involved in the first Hydration Sphere (HS) of Atra are given.

From P1 to P2, the number of water molecules in $HS_{Atra}$ varies from 14 to 32, with a mean value of [21.9;24.1]. Atra hydration state for these singularities is linked with small θ values (see $\bar{\theta }$ in Table 1). Moreover, the water molecule(s) that hydrate(s) Ca1 are included into $HS_{Atra}$, thus the complex Atra-Ca1 has the same hydration state as Atra. In Figure 5, snapshots representative of each singularity are depicted. They were chosen to reflect the mean θ angle and mean Atra and cation hydration states as much as possible.

For P1, $\bar{\theta }$ is low (7.1^∘^), thus, Atra is almost parallel to the surface. The chosen P1 snapshot shows an angle of 6.1^∘^, Atra parallel to the surface bonds to Ca1 within a monodentate manner *via* N1 (Ca1−N1=2.411 Å). The water molecules of the Atra hydration sphere, $HS_{Atra}$ (=20 in P1 snapshot), are distributed around the pesticide but not between the surface and Atra, conserving the Atra-surf dispersion effects. Both alkyl groups, ethyl and isopropyl, lie at average distances to the surface similar to the reaction coordinate ξ (see Figure 6). The Ca1 hydration sphere in P1 includes one to two molecules; one being predominant (see Figure 7). At the same time, $HS_{Ca2}$ varies from four to six water molecules; four being preferred (see Figure 8).

For M1, the situation is quite similar to P1 with $\bar{\theta }$ slightly larger (13.1^∘^) and thus, with $HS_{Atra}$ containing on average two more water molecules. In contrast to P1, Ce–surf and Ci–surf do not follow the same tendency: Ci–surf remains close to ξ, whereas Ce–surf becomes longer reducing the dispersion effects with the surface but allowing the entrance of water molecules in the space thus created (see Figure 6). In the M1 snapshot, we can observe that the monodentate interaction site between Atra and Ca1 is maintained (Ca1−N1=2.683 Å). Concerning $HS_{Ca1}$ and $HS_{Ca2}$, more hydration states are accessible than in P1: zero to two, and two to six, respectively. The most frequent combination is $HS_{Ca1}=1$ and $HS_{Ca2}=5$ (see Figure 7 and Figure 8).

In P2, the θ angle keeps on increasing ($\bar{\theta }=20.8^{\circ }$). Ci–surf distance distribution stays similar to M1. The dispersion interaction between the largest alkyl group, i.e., isopropyl, and the surface stands still. On the other hand, the ethyl group continues to move away from the surface (see Figure 6). Atra interacts with Ca1 mostly in a bidentate way involving N1 and Cl. In the P2 snapshot, Ca1−N1=2.548 Å and Ca1−Cl=3.132 Å. Let us notice that in the gas phase at B3LYP/6-31G*, those distances are slightly shorter, i.e., Ca1−N1=2.354 Å and Ca1−Cl=2.878 Å [12,13]. The mean hydration of the pesticide is similar to M1, as well as the accessible $HS_{Ca1}$, whereas, $HS_{Ca2}$ increases from three to seven. The P2 snapshot shows the most frequent combination, namely $HS_{Ca1}=1$ and $HS_{Ca2}=6$, but $HS_{Ca1}=2$ and $HS_{Ca2}=5$ is almost equiprobable (see Appendix A).

The biggest changes in inclination and hydration of Atra occur when moving from P2 to M2 corresponding to the largest free energy barrier of the profile (11.5 kcal·mol^−1^). On average, six more water molecules enter the first HS of Atra (see Table 1). At the same time, $\bar{\theta }$ undergoes its biggest increase ever: $+27.9^{\circ }$ from 20.8 to $48.7^{\circ }$. Thus, Atra clearly quits its parallel orientation to the surface. Moreover, none of the alkyl groups conserve dispersion interactions with the surface. The angle increase and the loss of dispersion effects are obviously correlated and contribute to the high free energy barrier. The growth of $HS_{Atra}$ is not sufficient to compensate the main Atra–surf dispersion effects, which are lost. Only the Cl/surf dispersion remains, Cl–surf distance being always shorter than the reaction coordinate. The distribution presents two peaks centered on 2.4 Å and 3.6 Å. The chosen M2 snapshot shows 29 water molecules in $HS_{Atra}$, the clear remoteness of the alkyl groups as well as the Cl atom small distance to the surface with a θ value of 55.19^∘^. The Ca1 hydration sphere in M2 includes zero to three molecules, with three being predominant (see Figure 7). At the same time, $HS_{Ca2}$ varies from one to four water molecules, with three being preferred (see Figure 8).

As a result of the small free energy difference between M2 and P3 (0.5 kcal·mol^−1^>), their descriptions are similar. When M2 presents several peaks with a normalized angle and distance distributions, in P3, only the most intense of those peaks is conserved (noted in bold for M2). For P3, the normalized distribution of θ is centered on 58^∘^ (M2 peaks around 31^∘^ and **59^∘^**). The greater the θ angle, the shorter the Cl–surf distance. Thus, in P3, the Cl–surf distance distribution is centered on 2.6 Å (Cl–surf M2 peaks around **2.4 Å** and 3.6 Å). For those singularities, the shortest Cl–surf distances are found. Concerning the distributions of the alkyl group distance to the surface, Ce–surf and Ci–surf are centered on 6.8 Å and 6.6 Å, respectively (Ce–surf M2 peaks around 5.7 Å and **6.7 Å**; Ci–surf M2 peaks around 5.5 Å and **6.5 Å**). Concerning the cation hydration spheres, Ca1 and Ca2 are both surrounded by three water molecules. At the same time, $HS_{Atra}$ varies from 24 to 35 for M2 and from 24 to 38 in P3, resulting in comparable mean values: 30.3 and 31.7, respectively. Finally, M2 resembles P3 more closely than its upstream minimum P2.

After P3, the free energy keeps on increasing without any clear singularity. To follow the evolution of the system when ξ is increased, three zones centered around 6.0, 7.0 and 8.0 Å were defined. For these zones, the majority of results are presented on Figure 3, Figure 4 and Figure 6 and on Table 1 and Table 2. Cation–surf distance normalized distributions are presented in Appendix A and the snapshots are depicted in Appendix A. The last interaction between Atra and the surface is through the Cl atom. This dispersion interaction is finally lost when Cl–surf neighbors 5.3 Å. In the 6.0 Å and 7.0 Å zones, there are structures that do or do not have this interaction, whereas in 8.0 Å zone, all the structures present long Cl–surf distances (>5.3 Å), no dispersion interaction between Atra and surf remains. The chosen snapshots (see Appendix A) shows this observation: in 6.0 Å and 7.0 Å, no water molecule lies in between the Cl atom of the pesticide and the Mont surface; for 8.0 Å, four water molecules enter the space between Cl and the surface. From P3, the Atra hydration sphere is growing but seems to reach a plateau after 7.0 Å with $HS_{Atra}\approx 36$. In those cases, as Atra is going further to the surface, less water molecules are shared between Ca1 and Atra: 1.9 (over $HS_{Ca1}=3.0$) around 7.0 Å and 1.1 (over $HS_{Ca1}=3.3$) around 8.0 Å. Here, the maximum of $HS_{Atra}$ includes 44 water molecules. As seen previously, a link can be made between θ angle distribution and Cl–surf distance distribution. In 6.0 Å, Cl–surf distribution presents two broad peaks: the first and most intense one centered on 3.2 Å corresponds to no water molecules between Cl and surf, simultaneously, the θ angle ranges from 25 to 69^∘^; the second peak, with clearly less intensity, centered on 6.2 Å, gathers structures with more hydrated Cl, thus, the θ angle varies between −20 and $+20^{\circ }$. Around 7.0 Å, the same tendency is observed but the peaks are closer for Cl–surf distance and θ angle distributions. When the Atra/surf interaction is lost for 8.0 Å, longer Cl–surf distances are accessible ([5.3;7.8] Å). The corresponding angles range from $0^{\circ }$ to $55^{\circ }$ with a center value for this peak around $25^{\circ }$. A greater hydration of the Cl atom of the pesticide implies a smaller $\bar{\theta }$ value: $27.1^{\circ }$ among the three last zones.

For all singularities and the three aforementioned zones, Ca1–Ca2 normalized distributions are presented in Appendix A, the corresponding mean Ca1–Ca2 distances are given in Table 1. As a reference, the Ca1–Ca2 distance in dry Mont is 10.341 Å [14,15]. In each case, the mean Ca1–Ca2 distance is shorter than or equal to the reference value. The greater the screening of the cation charge, the smaller the distance between Ca1 and Ca2. Screening is due to the presence of water molecules around both cations, but for Ca1, the pesticide also has a screening influence. The smallest distances are obtained for P1, M1 and P2 because of the Atra/Ca1 interaction. In M2 and P3, the interaction between Atra and Ca1 is lost. Despite larger $HS_{Ca1}$ (2.1 in M2 and 3.0 in P3), the screening of Ca1 charge is weaker due to the loss of Atra/Ca1 interaction. Therefore, Ca1–Ca2 distances are larger than in P1, M1 and P2 but of the same magnitude as in dry Mont.

## 3. Materials and Methods

### 3.1. Montmorillonite Model

Montmorillonite is a 2:1 phyllosilicate containing one octahetral (O) sheet sandwiched between two tetrahedral (T) sheets. Oxygen (O^2−^) and hydroxyl (OH^−^) are located at the apices of the T and O polyhedrons. This dioctahedral smectite (T/O/T) has Si^4+^ cations in all sites of the T sheets and Al^3+^ in two thirds of the octahedral sites. Around 16.7% of the isomorphic substitutions of Al^3+^ by Mg^2+^ are distributed within the O layers. This distribution results in a net negative structural charge of the clay, which is balanced by adsorbed Ca^2+^ ions.

Unit cell parameters of pyrophyllite (a, b, c, α, β, γ) were found in good agreement with X-ray experimental data (R. Wardle et al. [16]) after being optimized at PBE-D2 level (see previous works [14,15,17]). Since the structure of pyrophyllite is similar to montmorillonite but without an isomorphic substitution nor a compensating cation in the interlayer spacing, it allowed us to build our montmorillonite model. The simulation cell consists of six unit cells (3a×2b×c). The triclinic cell parameters are kept fixed during the Molecular Dynamics (MD) simulations at the following values: 15.500 Å × 17.931 Å ×25.000 Å and $\alpha =91.030^{\circ }$, $\beta =100.374^{\circ }$, $\gamma =89.757^{\circ }$. The basal spacing $d_{001}$ is equal to 24.6 Å$\left(d_{001}=c\sin\beta \right)$ resulting in an interlayer spacing δ≈18 Å (see Figure 9).

### 3.2. Molecular Dynamics Simulations

MD simulations were performed using version 3.15.1 of CPMD software [18] within the Car-Parrinello framework [19] and a Density Functional Theory (DFT) potential using a generalized gradient approximation functional, PBE [20,21,22,23]. Grimme’s D2 corrections were selected to accurately model dispersion interactions [24,25,26]. The simulation cell was periodically replicated in 3D. Troullier-Martins norm-conserving pseudopotentials [27,28] with a plane-wave basis set describe valence-core interactions. A kinetic energy cutoff of 80 Ry for the plane-wave basis with a 4.0 a.u. time step was employed to integrate the equations of motion. Owing to the large computational cell, the Brillouin zone sampling was restricted to the Γ point using the Monkhorst-Pack method [29]. Considering the large size of the simulation cell, the default option for the Ewald term was used in the CPMD software. The fictitious mass of the electronic wavefunction was set to 500 a.u. The calculations were performed in a canonical (NVT) ensemble applying the Langevin thermostat procedure [30] at the simulation temperature of 350 K to avoid the over-structuration of water using PBE compared to experiment [31,32,33]. One hundred and thirty-two water molecules were present in the interlayer space in all the trajectories of this work.

### 3.3. Potential of Mean Force

Nineteen biased Car-Parrinello MD simulations were performed using one collective variable being our reaction coordinate noted ξ (see Table 3). The reaction coordinate ξ was defined as the distance, along the *z*-axis of the simulation cell, between the location of the center of mass of atrazine and the surface (surf) containing the basal oxygen atoms, Ob (see Figure 9). An umbrella sampling method [34,35,36] was used to observe the desorption of atrazine from the hydrated montmorillonite surface. Biased harmonic potentials centered on $\xi_{i}$ values were added to the system using PLUMED, an open-source community-developed library [37,38,39]. For a given $\xi_{i}$, after an equilibration time of 4.0 ps, we performed NVT simulations lasting from 12 to 16 ps resulting in a cumulative simulation time of 209 ps.

Spanning the range from 3.583 Å to 8.920 Å for the reaction coordinate ξ allowed to move atrazine from a location close to the surface to the middle of the interlayer spacing. The sampling interval changed according to the free energy profile singularities (from 0.029 Å minimum to 0.529 Å maximum) in order to obtain satisfying overlapping distributions of ξ (see Appendix A).

The Potential of Mean Force (PMF) was derived to quantify the free energy *F* of the atrazine desorption from hydrated montmorillonite surface. The changes in *F* were due to the changes in the system configurations and were described by the PMF [40,41,42,43]. In an NVT statistical ensemble, the free energy *F* of a system in thermodynamic equilibrium could be written as: (1)$$F=-k_{B}TlnZ$$
*Z* being the canonical partition function, *T*, the temperature (K) and $k_{B}$, the Boltzmann constant (J·K^−1^).

For each *i* window, the dynamic simulation is biased introducing into the Hamiltonian of the system, a harmonic potential *V* centered on the $\xi_{i}$ value: (2)$$V=\frac{1}{2}k_{i}{\left(\xi -\xi_{i}\right)}^{2}$$
$k_{i}$ is the harmonic constant (see Table 3) for window *i*. The values of the $k_{i}$ were selected to obtain the best sampling of the potential energy surface (PES).

For the hydrated interacting system under study (Atra;Mont), a free energy curve was obtained for each window *i* noted W(ξ) being the PMF and expressed by the following equation: (3)$$W\left(\xi \right)=-k_{B}Tln\left(g\left(\xi \right)\right)$$
g(ξ) was the distribution function of distances for the pair (Atra; Mont).

The entire free energy curve was rebuilt from the ξ distributions of each biased MD trajectory *via* the WHAM self consistent method (Weighted Histogram Approximation Method) [41,44,45]. The WHAM parameters were set to 1000 for the number of bins and to $10^{-6}$ kcal·mol^−1^ for the precision on final free energy. Thus, the number of points in the free energy curve is 1000 and the WHAM iterative resolution process is considered to be converged when no free energy value for any simulation window changes by more than $10^{-6}$ kcal·mol^−1^ on consecutive iterations.

## 4. Conclusions

In the beginning of the desorption process, Atra interacts with the surface through Ca1. The first two minima, P1 and P2, are free energetically quasi degenerated. In P1, Atra bonds to Ca1 via the monodentate site N1, whereas in P2, both N1 and Cl are involved in the interaction resulting in a bidentate binding site. In P1, Atra is parallel to the Mont surface maximizing the Atra/surf dispersion interactions. Moving to P2, Atra changes its orientation from parallel to inclined. The M1 barrier connecting P1 to P2 is low in free energy (≈0.5 kcal·mol^−1^) and corresponds to the remoteness of the ethyl group with respect to the surface and thus, reducing Atra/surf dispersion effects. Between P2 and M2, the barrier is clearly higher (≈11.5 kcal·mol^−1^). This is due to the loss of isopropyl/surf interaction, which is mainly dispersive, for Atra only the Cl/surf dispersion remains. M2 resembles more P3 than its upstream minimum P2. From P3, when ξ keeps on increasing, the free energy also increases without any clear singularity. Finally, around ξ=8.0 Å, the last Atra/surf dispersion interaction via the Cl atom vanishes and Atra is completely desorbed. The mechanism evidenced in the desorption process is the one presented in Figure 9a: Atra desorbed alone and Ca1 remaining on the Mont surface, which is similar to the previously studied fenhexamid (Fen) pesticide desorption from the same clay surface [11]. However, the Atra desorption profile differs from the Fen desorption profile. First, the highest barrier to desorb Fen from Mont is ≈6.5 kcal·mol^−1^, which is about half the barrier to desorb Atra. Moreover in the case of Fen, the desorption is driven by both cation hydration, whereas here, the process is led by the dispersion effects of the Atra alkyl groups. As the mechanism is driven by atrazine alkyl groups and their sizes, we can deduce the desorption behavior of the various atrazine metabolites. Therefore, we project an easier desorption for deisopropylatrazine, than atrazine or deethylatrazine. It would be interesting to confirm this hypothesis by simulating the desorption processes of Atra degradation products.

## Figures and Tables

**Figure 1 ijms-25-01604-f001:**
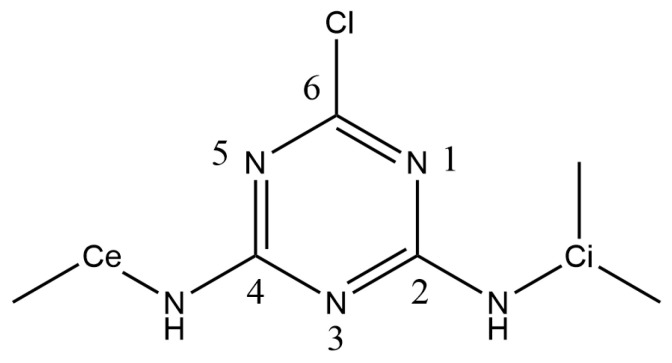
Structural formula of the atrazine herbicide, C_8_H_14_ClN_5_, 2-isopropylamino-4-ethylamino-6-chloro-1,3,5-triazine, and numbering of the atoms. Ce is the label for the ethyl carbon atom linked to the amino group and Ci is the label for the isopropyl carbon atom linked to the amino group.

**Figure 2 ijms-25-01604-f002:**
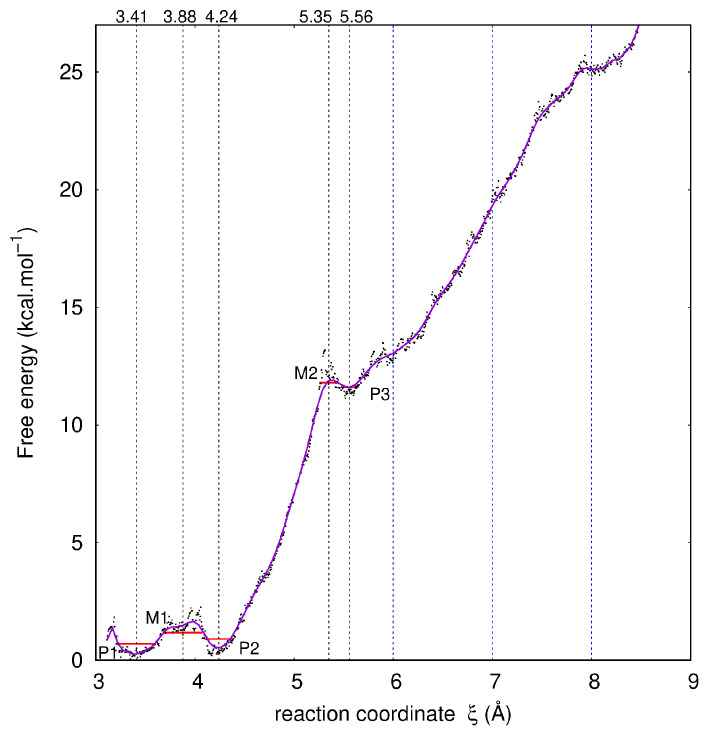
Free energy profile of atrazine desorption from the montmorillonite surface as a function of ξ, the distance from the center of mass of atrazine to the surface containing the oxygen atoms Ob. Every black dot represents a structure visited during the 19 MD trajectories. The continuous purple line gives the free energy profile of the desorption after smoothing. P1, P2 and P3 are the minima or energetic wells of this profile. M1 and M2 are the maxima. The zones delimited by the red horizontal lines enclose the structures considered as belonging to the minima and to the maxima. Numbers on the top of the graph are the ξ values for each singularity.

**Figure 3 ijms-25-01604-f003:**
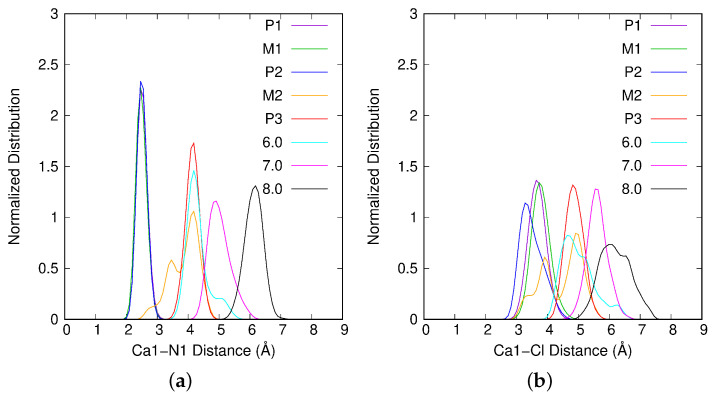
(**a**) Ca1-N1 and (**b**) Ca1-Cl normalized distance distributions for each of the singularities P1, M1, P2, M2 and P3 and also for the ξ values 6.0, 7.0 and 8.0 Å.

**Figure 4 ijms-25-01604-f004:**
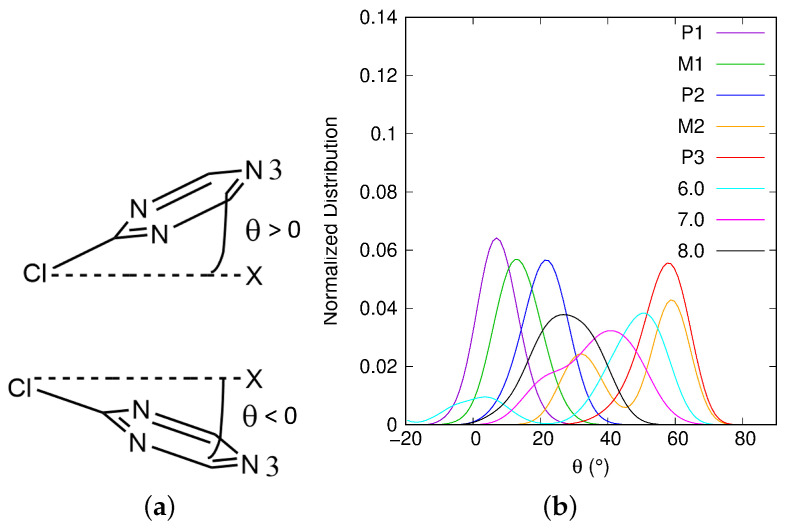
(**a**) θ angle definition: top view, θ>0; bottom view, θ<0. (**b**) θ normalized distributions for each of the singularities P1, M1, P2, M2 and P3 and also for the ξ values 6.0, 7.0 and 8.0 Å. θ is given in degrees.

**Figure 5 ijms-25-01604-f005:**
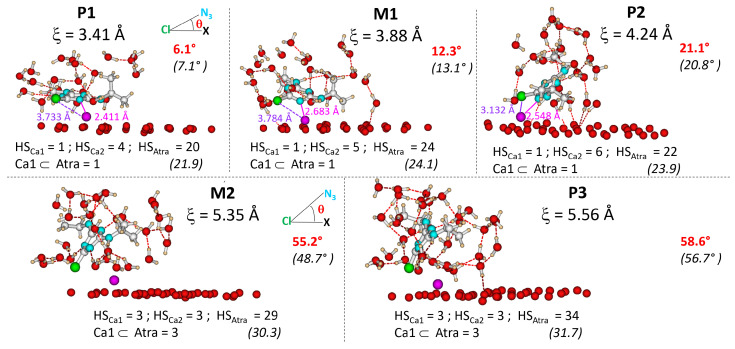
Snapshots taken in the five singularities P1, M1, P2, M2 and P3 of the free energy profile and their ξ values. For a sake of clarity, only the Ob oxygen atoms are represented for the Mont surface. For the same reason, only the water molecules involved in the first HS of both atrazine and Ca1 are represented. $HS_{Ca1}$, $HS_{Ca2}$ and $HS_{Atra}$ the number of water molecules involved in the first HS of each entity are noticed. In red, the θ angle is given for the snapshot. In parentheses, the mean values of $HS_{Atra}$ and θ are also given. Atom color codes: H, orangy beige; C, gray; N, blue; O, red; Cl, green; Ca, magenta.

**Figure 6 ijms-25-01604-f006:**
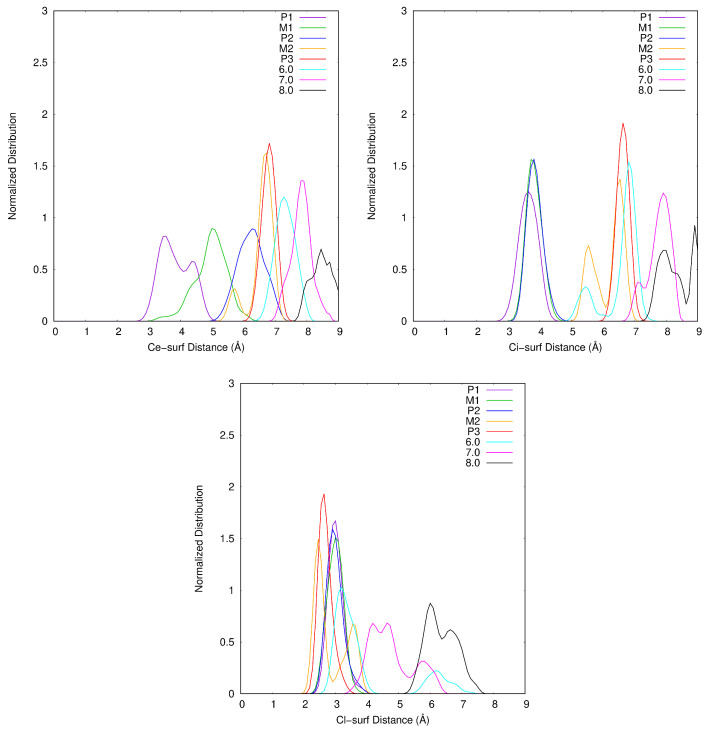
Ce–surf, Ci–surf and Cl–surf normalized distance distributions for each of the singularities P1, M1, P2, M2 and P3 of the free energy profile and also for the ξ values 6.0, 7.0 and 8.0 Å.

**Figure 7 ijms-25-01604-f007:**
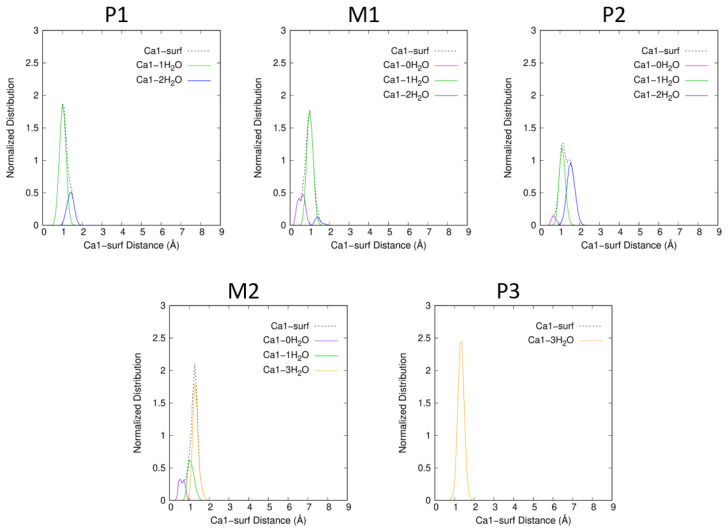
Ca1–surf normalized distance distributions for each of the singularities P1, M1, P2, M2 and P3 of the free energy profile. The structures were sorted according to the number of water molecules in the first HS of Ca1 ($d_{Oe-Ca1}\leq 3.0$ Å).

**Figure 8 ijms-25-01604-f008:**
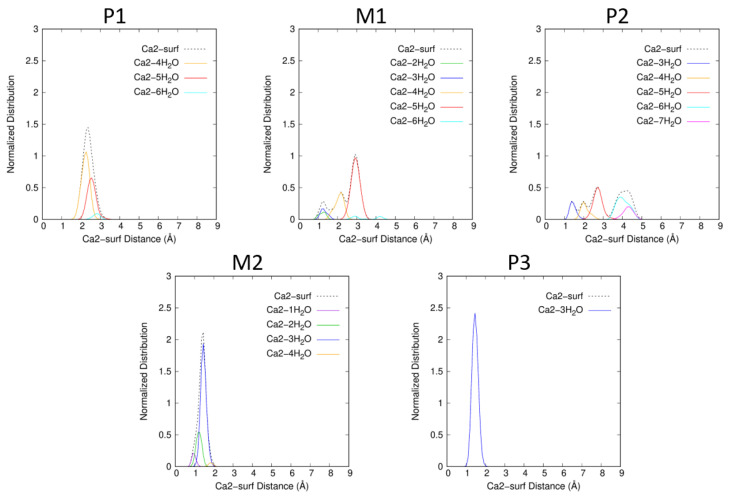
Ca2–surf normalized distance distributions for each of the singularities P1, M1, P2, M2 and P3 of the free energy profile. The structures were sorted according to the number of water molecules in the first HS of Ca2 ($d_{Oe-Ca2}\leq 3.0$ Å).

**Figure 9 ijms-25-01604-f009:**
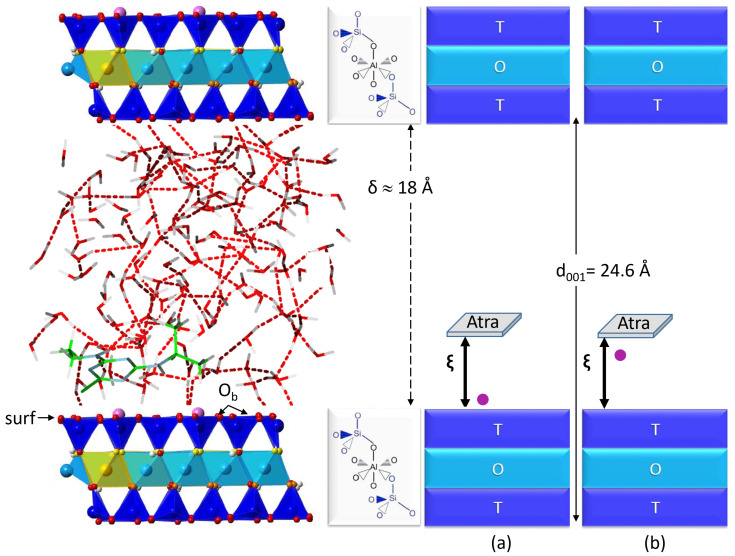
Ca^2+^–Montmorillonite (Mont) computational cell with the atrazine and 132 water molecules in the interlayer spacing. Mont contains four substitutions of Al^3+^ (light blue) by Mg^2+^(yellow) in the octahedral (O) sheet and two Ca^2+^ cations (magenta) compensate the net negative charge carried by the clay layer and are situated above substitutions (ditrigonal ring of basal oxygen atoms, Ob). Ca1 is the Ca^2+^ interacting with Atra and Ca2 does not interact with Atra. The whole set of Ob atoms forms the Mont surface, noted surf. Si^4+^ ions in the tetrahedral (T) sheet are in dark blue. δ designates the interlayer spacing and $d_{001}$, the basal spacing. On the right, mechanism propositions for the desorption of Atra from Mont surface: (**a**) Atra alone is desorbed from Mont, Ca1 remains attached to Mont; (**b**) Atra–Ca1 complex is desorbed from Mont.

**Table 1 ijms-25-01604-t001:** The mean number of water molecules involved in the first hydration spheres of Ca1 ($HS_{Ca1}$), Ca2 ($HS_{Ca2}$), Atrazine ($HS_{Atra}$) and Atrazine–Ca1 entity ($HS_{Atra\text{-}Ca1}$) for each of the singularities P1, M1, P2, M2 and P3 and also for around the ξ values 6.0, 7.0 and 8.0 Å. $HS_{tot}$ is the mean number of water molecules around Atrazine, Ca1 and Ca2. Mean Ca1–Ca2 distances are also reported in Å, as well as the mean θ angle, noted $\bar{\theta }$ and given in degrees. When $HS_{Ca1}$ is included into $HS_{Atra}$, it is written in bold.

	${\mathrm{HS}}_{\mathrm{tot}}$	${\mathrm{HS}}_{\mathrm{Atra}\text{-}\mathrm{Ca}1}$	Ca1 ⊂ Atra ^a^	${\mathrm{HS}}_{\mathrm{Atra}}$	${\mathrm{HS}}_{\mathrm{Ca}1}$	${\mathrm{HS}}_{\mathrm{Ca}2}$	Ca1–Ca2 (Å)	$\bar{\theta }$ (**^∘^**)
P1	26.4	**21.9**	**1.2**	**21.9**	**1.2**	4.5	9.10	7.1
M1	28.6	**24.1**	**0.8**	**24.1**	**0.8**	4.5	8.57	13.1
P2	29.2	**23.9**	**1.4**	**23.9**	**1.4**	5.2	9.74	20.8
M2	33.1	30.4	2.1	30.3	2.1	2.7	10.34	48.7
P3	34.9	31.9	2.8	31.7	3.0	3.0	10.29	56.7
6.0 Å	37.8	33.6	2.7	33.3	3.0	4.2	10.33	40.3
7.0 Å	41.6	37.7	1.9	36.6	3.0	3.9	9.70	36.5
8.0 Å	41.6	38.1	1.1	35.8	3.3	3.5	10.00	27.1

^a^ Number of water molecules in common between $HS_{Atra}$ and $HS_{Ca1}$.

**Table 2 ijms-25-01604-t002:** The minimum (MIN), maximum (MAX) and mean (MEAN) numbers of water molecules involved in the first HS of Atrazine for each of the singularities P1, M1, P2, M2 and P3 and also around the ξ values 6.0, 7.0 and 8.0 Å.

	MIN	MAX	MEAN
P1	14	29	21.9
M1	15	32	24.1
P2	16	30	23.9
M2	24	35	30.3
P3	24	38	31.7
6.0 Å	25	41	33.3
7.0 Å	30	44	36.6
8.0 Å	26	43	35.8

**Table 3 ijms-25-01604-t003:** Parameters of the 19 biased MD trajectories ^a^.

*i*	$\xi_{i}$ (Å)	$k_{i}$ ($\mathrm{kcal}\cdot {\mathrm{mol}}^{-1}\cdot$Å^−2^)	$t_{i}$ (ps)
1	3.583	28.6	13
2	3.628	14.3	16
3	3.805	57.1	4
4	3.805	28.6	9
5	4.157	14.3	12
6	4.486	14.3	14
7	5.216	14.3	12
8	5.421	57.1	5
9	5.421	28.6	7
10	5.615	28.6	13
11	5.745	14.3	14
12	6.274	14.3	12
13	6.803	14.3	14
14	7.332	14.3	14
15	7.832	57.1	4
16	7.832	28.6	10
17	7.861	14.3	12
18	8.391	14.3	12
19	8.920	14.3	12

^a^ *i* is the window index, $\xi_{i}$ the value of the reaction coordinate around which the harmonic potential oscillated, $k_{i}$ is the associated harmonic constant, and $t_{i}$ the duration of the simulation (ps).

## Data Availability

Additional data are already joined in Appendix A.

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
