# Peer review of "Atrazine Desorption Mechanism from an Hydrated Calcium Montmorillonite—A DFT Molecular Dynamics Study"

_ijms, 2024, doi:10.3390/ijms25031604_

Round 1

Reviewer 1 Report

Comments and Suggestions for Authors

The presented paper “Atrazine Desorption Mechanism from an Hydrated Calcium Montmorillonite - A DFT Molecular Dynamics Study” studies potentially toxic component desorption from clay material, with no doubt the subject is interesting and important. The problem is carefully studied using advanced theoretical methods. Nevertheless, only one system of potential interest is presented, therefore the overall merit of the paper is rather average / low in my opinion. For such reason I think the Editor should decide if the manuscript can be published in IJMS.

The methods are clearly presented and correctly used, the data are carefully analysed and the results are consistent. The only weakness of the manuscript (in scientific terms) is that the simulations were performed for one particular system only. The main and open question is how the results would compare with other atrazine derivatives desorption from the same material? Therefore I would suggest to repeat the same computational procedure for at least one other atrazine derivative.

The other, minor question is about the simulation temperature. Could the authors explain why 350K has been chosen? What implication to simulation quality (if any) it has?  How it affect the energy barriers and how relevant it is for experimental studies. Why the room temperature was not used in the simulations?

Comments on the Quality of English Language

In general the Quality of language is good, nevertheless some corrections are required. I would suggest careful reading of the paper and some style corrections. There are numerous points where the style could be improved (too many to list).

For example the line 60 "D. Tunega and coworkers published a paper about the stability of atrazine–smectite intercalates using DFT simulations and implementing experimental data" - it suggest that Tuenga used DFT to publish a paper, while of course the authors meant that the results published by Tuenga were obtained using DFT simulation data.

Some grammar corrections are also required, for example line 56 " confirme or complete" -> should be "confirm or complete"

Author Response

Comments and Suggestions for Authors

The presented paper “Atrazine Desorption Mechanism from an Hydrated Calcium Montmorillonite - A DFT Molecular Dynamics Study” studies potentially toxic component desorption from clay material, with no doubt the subject is interesting and important. The problem is carefully studied using advanced theoretical methods. Nevertheless, only one system of potential interest is presented, therefore the overall merit of the paper is rather average / low in my opinion. For such reason I think the Editor should decide if the manuscript can be published in IJMS.

The methods are clearly presented and correctly used, the data are carefully analysed and the results are consistent. The only weakness of the manuscript (in scientific terms) is that the simulations were performed for one particular system only. The main and open question is how the results would compare with other atrazine derivatives desorption from the same material? Therefore, I would suggest to repeat the same computational procedure for at least one other atrazine derivative.

One of the other reviewers points out that: “the use of enhanced sampling (umbrella sampling) techniques requires large amounts of computer time and also careful, patient post-processing analyses”. Indeed, to obtain the free energy profile of desorption of atrazine from the montmorillonite surface about 300 000 hours of CPU time were necessary, which corresponds to more than 500 days of real time to obtain the results presented here. Moreover, post-processing analyses were also time demanding. Thus, studying desorption of the atrazine derivatives is of course of a great interest but deserve a dedicated study, which is planned in our group as mentioned on L334-336.

In addition, regarding the large amount of CPU time necessary to converge such a desorption curve, we project to develop a less demanding potential perhaps using IA. This type of potential is on development in our group. We plan to use such less costly potential to study atrazine derivatives.

The other, minor question is about the simulation temperature. Could the authors explain why 350K has been chosen? What implication to simulation quality (if any) it has?  How it affects the energy barriers and how relevant it is for experimental studies. Why was the room temperature not used in the simulations?

The PBE functional with or without dispersion correction, tends to overstructure water compared to experiment. The thermostat of the simulations is set to 350K instead of 298K (ambient temperature) to try to overcome this over structuration issue and get intermolecular first neighbor distances as close as possible to 2.80Å, which is the most recent and accurate experimental value at ambient temperature. On L103-106, this issue is addressed, and some references are given. The liquid phase of water was confirmed all along the desorption process following the radial distribution function of Oe-Oe distance (Oe oxygen atom of water molecules, see L155 and Supporting information).

Comments on the Quality of English Language

In general, the Quality of language is good, nevertheless some corrections are required. I would suggest careful reading of the paper and some style corrections. There are numerous points where the style could be improved (too many to list).

For example, the line 60 "D. Tunega and coworkers published a paper about the stability of atrazine–smectite intercalates using DFT simulations and implementing experimental data" - it suggests that Tunega used DFT to publish a paper, while of course the authors meant that the results published by Tunega were obtained using DFT simulation data.

Some grammar corrections are also required, for example line 56 " confirme or complete" -> should be "confirm or complete".

The errors pointed out by the referee have been corrected and a careful reading has been made to detect any other language errors.

Reviewer 2 Report

Comments and Suggestions for Authors

The manuscript on “Atrazine Desorption Mechanism from an Hydrated Calcium Montmorillonite - A DFT Molecular Dynamics Study” by Q. Desdion, F. Bessac and S. Hoyau is a well-prepared and competently carried out theoretical project. The topic is of quite high importance, since the title herbicide is a persistent molecule, still circulating in the environment. The use of enhanced sampling (umbrella sampling) techniques requires large amounts of computer time and also careful, patient post-processing analysis. The design of the models is adequate and related to the real-life scenario of adsorption in clay minerals. The results are well described, and I do not find any serious scientific flaws, therefore I recommend publication of the manuscript in almost unchanged form. I would only like to ask – for completeness – about adding CPMD version number at line 93, and explicit information (close to line 96) whether the periodicity was 3D (as I suppose) or rather only 2D (slab), and whether any Ewald summation (TESR keyword of CPMD) was employed. Similarly, the procedure of WHAM reconstruction of the free energy profile, including smoothing, should be described close to line 140 (if there are any modifiable parameters in the process): please note that the M2 singularity (see Figure 3) is very similar in energy to the P3 minimum, and the Figure 3 suggests that tiny differences in smoothing could result in large changes in the barrier.

Author Response

Comments and Suggestions for Authors

The manuscript on “Atrazine Desorption Mechanism from an Hydrated Calcium Montmorillonite - A DFT Molecular Dynamics Study” by Q. Desdion, F. Bessac and S. Hoyau is a well-prepared and competently carried out theoretical project. The topic is of quite high importance, since the title herbicide is a persistent molecule, still circulating in the environment. The use of enhanced sampling (umbrella sampling) techniques requires large amounts of computer time and also careful, patient post-processing analysis. The design of the models is adequate and related to the real-life scenario of adsorption in clay minerals. The results are well described, and I do not find any serious scientific flaws, therefore I recommend publication of the manuscript in almost unchanged form.

I would only like to ask – for completeness – about adding CPMD version number at line 93, and explicit information (close to line 96) whether the periodicity was 3D (as I suppose) or rather only 2D (slab), and whether any Ewald summation (TESR keyword of CPMD) was employed.

The requested information has been added to the manuscript:

L93 CPMD version 3.15.1

L96 3D periodicity

L101 A sentence was added to explain that the default for the TSER keyword was used in the simulations due to the large size of the computational cell.

Similarly, the procedure of WHAM reconstruction of the free energy profile, including smoothing, should be described close to line 140 (if there are any modifiable parameters in the process): please note that the M2 singularity (see Figure 3) is very similar in energy to the P3 minimum, and the Figure 3 suggests that tiny differences in smoothing could result in large changes in the barrier.

The requested information has been added to the manuscript:

L142 WHAM parameters were detailed: nbins=1000, tol=10-6

L154 The smoothing was applicated after the WHAM process. The following sentence was added to the text: ”Taking the minimum free energy point for a well and the maximum free energy point for a maximum, these barriers are worth 2 and 13 kcal/mol, respectively.”

Round 2

Reviewer 1 Report

Comments and Suggestions for Authors

I'm glad the Authors have addressed my comments. I still think it would be useful to repeat the simulations for other Atra derivatives, nevertheless I fully understand the point of CPU costs... I hope this is one of the Authors ideas for future work. I think the paper is publishable now.